# The WASABI dataset: cultural, lyrics and audio analysis metadata about 2 million popular commercially released songs

Michel Buffa, Elena Cabrio, Michael Fell, Fabien Gandon, Alain Giboin,
Romain Hennequin, Franck Michel, Johan Pauwels, Guillaume Pellerin,
Maroua Tikat, and Marco Winckler

University Côte d'Azur, Inria, CNRS, I3S, France:
`michel.buffa@univ-cotedazur.fr`, `elena.cabrio@univ-cotedazur.fr`,
`fabien.gandon@inria.fr`, `alain.giboin@inria.fr`, `franck.michel@cnrs.fr`,
`maroua.tikat@univ-cotedazur.fr`, `winckler@univ-cotedazur.fr` .
Università degli Studi di Torino: `michaelkurt.fell@unito.it`.
Queen Mary University of London: `j.pauwels@qmul.ac.uk`.
IRCAM: `guillaume.pellerin@ircam.fr`.
Deezer Research: `rhennequin@deezer.com`

**Abstract.** Since 2017, the goal of the two-million song WASABI database
has been to build a knowledge graph linking collected metadata (artists,
discography, producers, dates, etc.) with metadata generated by the anal-
ysis of both the songs' lyrics (topics, places, emotions, structure, etc.)
and audio signal (chords, sound, etc.). It relies on natural language pro-
cessing and machine learning methods for extraction, and semantic Web
frameworks for representation and integration. It describes more than
2 millions commercial songs, 200K albums and 77K artists. It can be
exploited by music search engines, music professionals (e.g. journalists,
radio presenters, music teachers) or scientists willing to analyze popu-
lar music published since 1950. It is available under an open license, in
multiple formats and with online and open source services including an
interactive navigator, a REST API and a SPARQL endpoint.

**Keywords:** music metadata, lyrics analysis, named entities, linked data

## 1 Introduction

Today, many music streaming services (such as Deezer, Spotify or Apple Music)
leverage rich metadata (artist's biography, genre, lyrics, etc.) to enrich listening
experience and perform recommendations. Likewise, journalists or archivists ex-
ploit various data sources to prepare TV/radio shows or music-related articles.
Music and sound engineering schools use these same data to illustrate and ex-
plain the audio production techniques and the history or music theory behind
a song. Finally, musicologists may look for hidden relationships between artists
(e.g influences, indirect collaborations) to support a claim. All these scenarios

have in common that they show the need for more accurate, larger and better linked music knowledge bases, along with tools to explore and exploit them.

Since 2017, the WASABI research project[1] has built a dataset covering more than 2M songs (mainly pop/rock and dub) in different languages, 200K albums and 77K artists. Musicologists, archivists from Radio-France, music schools and music composers also collaborated. While cultural data were collected from a large number of data sources, we also processed the song lyrics and performed audio analyses, enriching the corpus with various computed metadata addressing questions such as: What do the lyrics talk about? Which emotions do they convey? What is their structure? What chords are present in the song? What is the tempo, average volume, etc.? We partnered with the Queen Mary University of London (QMUL) and the FAST project[2] for extracting chords from the song audio, and linked to IRCAM's TimeSide[3] audio analysis API which for audio processings (beat detection, loudness, etc.). We deployed REST and SPARQL endpoints for requests and a GUI for exploring the dataset [6]. The dataset, Machine Learning models and processing pipeline are described and available[4] under an open license.[5]

Section 2 presents the context of the WASABI project and related works. In section 3, we explain the way we collected and processed data to build the corpus. Section 4 focuses on the formalization, generation and publication of the RDF knowledge graph. Section 5 presents several tools and visualizations built on top of the dataset and services. Finally, section 6 discusses quality assessment concerns while sections 7 and 8 discuss future applications and potential impact of the dataset and conclude with some perspectives.

## 2   State of the art and related work

There are large datasets of royalty-free music such as Jamendo (often used [5,26]) or others found in the DBTunes link directory, but we focus on the ones that cover commercial popular music (see Table 1) and we will see that few propose metadata on cultural aspects, lyrics and audio altogether. MusicBrainz offers a large set of cultural metadata but nothing about lyrics, for example. The Last.fm dataset contains tags that were used by some researchers for computing moods and emotions [13,7]. AcousticBrainz, a public, crowd-sourced dataset, contains metadata about audio and has been used by projects such as MusicWeb [2] and MusicLynx [3] to compute similarity models based on musical tonality, rhythm and timbre features.

The Centre for Digital Music of QMUL collaborated with the BBC on the use of Semantic Web technologies, and proposed music ontologies in several fields including audio effects and organology.

---

[1] Web Audio Semantic Aggregated in the Browser for Indexation, (Université Côte d'Azur, IRCAM, Deezer and Parisson) `http://wasabihome.i3s.unice.fr/`

[2] QMUL and the FAST project `http://www.semanticaudio.ac.uk/`

[3] TimeSide`https://github.com/Parisson/TimeSide`

[4] `https://github.com/micbuffa/WasabiDataset`

[5] Creative Commons Attribution-NonCommercial-ShareAlike 4.0 International (CC BY-NC-SA)

MusicLynx [3] provides an application to browse through music artists by exploiting connections between them, either extra-musical or tangential to music. It integrates open linked semantic metadata from various music recommendation and social media data sources as well as content-derived information. This project shares some ideas with the WASABI project but does not address the same scale of data, nor does it perform analysis on audio and lyrics content.

The Million Song Dataset project (MSD) processed a large set of commercial songs to extract metadata using audio content analysis [4], but did not take advantage of structured data (e.g. from DBpedia) to address uncertainties. Information such as group composition or orchestration can be very relevant to informing Music Information Retrieval (MIR) algorithms, but is only available in certain data sources (BBC, MusicBrainz, ...), and for many little-known artists this information is not available. It is here that the combination of audio and semantics finds its purpose, one reinforcing the other. The WASABI project provides a wider scope than the Million Song Dataset: it started as a challenge to build a datataset that would be twice as big with public domain development of open source tools and a richer cultural and lyric-related set of metadata.

The DOREMUS project [16] overlaps with WASABI but in a rather different context (classical and traditional music). DOREMUS performs the integration of MIDI resources (instead of MIR analysis), recommendation and automatic playlists generation. The WASABI ontology extends the Music Ontology (MO), yet the Performed Music Ontology[6] (part of LD4L) or DOREMUS ontology (based on FRBR) may be considered if future works need to model more accurately the differences between works, performances or expressions.

The Listening Experience Database (LED) collects people's music listening experiences as they are reported in documents like diaries or letters [1]. It mostly relates to legacy music that has little overlap with WASABI.

The MELD framework [21] supports the publication of musicology articles with multi-modal user interfaces that connect different forms of digital resources. Some development could be undertaken to allow musicologists publish articles that would leverage musical data from the WASABI RDF knowledge graph.

The MIDI Linked Data project [17] publishes a large set of MIDI files in RDF. Linked to DBpedia and relying on the Music Ontology, it could complement WASABI to jointly exploit MIDI files and audio and text analyses. Some MIDI content was used in WASABI during the evaluation of the chord extraction.

---

[6] `https://wiki.lyrasis.org/display/LD4P/Performed+Music+Ontology`

**Table 1.** Comparison with other datasets.

|  | Nb Songs | Linked Data | Audio analysis | Lyrics analysis | Cultural metadata | Type of music |
|---|---|---|---|---|---|---|
| WASABI | 2M | Yes | Yes | Yes | Yes | Commercial |
| MSD | 1M | No | Yes | Bag of words | Partial | Commercial |
| DOREMUS | 24k | Yes | No, Midi | Not relevant | Yes | Classical |
| MusicBrainz | 33M | Yes | No | No | Yes | Commercial |
| AcousticBrainz | 4M | Yes | Yes | No | MusicBrainz | Commercial |
| Jamendo | 200k+ | No | Chords | No | Yes | Royalty free |

## 3    Building the WASABI dataset

### 3.1    Assembling Cultural Data from Multiple Sources

One of the original goals of the WASABI project was to build a dataset comprising metadata produced by natural language processing applied to the lyrics. As shown in Figure 1, we therefore started from LyricsWikia, a wiki-based, crowd-sourced website gathering a large number of commercial song lyrics, metadata concerning the discography of thousands of artists (name, genre, labels, locations, duration, album release dates etc.). We collected data of 2M songs, 77K artists and 200K albums, including links and ids to songs, artists and albums on other platforms: Wikipedia, YouTube, MusicBrainz, Last.fm, Discogs, etc.

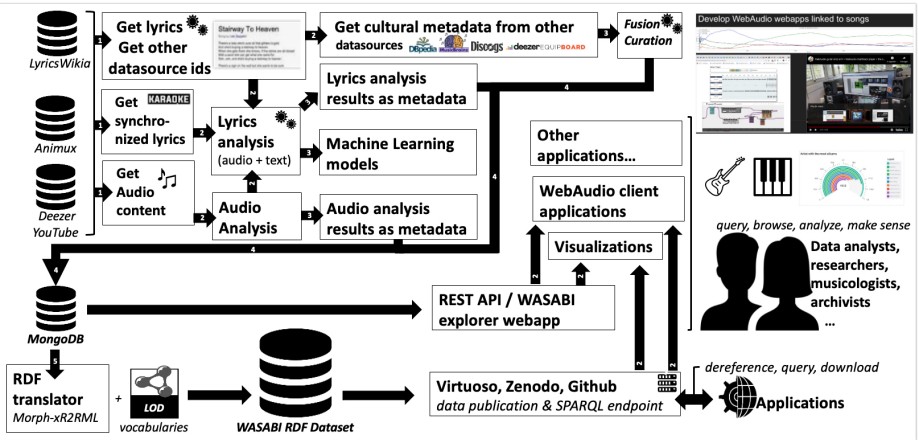

**Fig. 1.** WASABI pipeline from datasets and corpora to applications and end-users.

Subsequently, we used the links and ids to gather metadata from these multiple platforms. For instance, from several DBpedia language editions we retrieved music genres, awards and albums durations; from MusicBrainz: artist type (group, person, etc.), gender, life span, group members, albums' bar code,

**Table 2.** Ratios between external and those in our seed dataset LyricsWikia.

| Dataset | Songs | Artists | Albums | Comment |
|---|---|---|---|---|
| MusicBrainz | 57% | 78% | 45% | |
| DBpedia | 52% | 24% | 3.8% | |
| Deezer | 57% | 64% | 63% | 87% of the songs if we consider English songs only |
| Discogs | N/A | 69% | 41% | Only artists and discographies |
| Equipboard | N/A | 8.4% | N/A | Only famous artists/members had metadata |
| Final | 72% | 78% | 69% | Entries matched at least once in external sources |

release date, language; from Discogs: name variations (different ways to call the same song, album or artist) and artist real names that proved to be very relevant for consolidating the dataset; from EquipBoard: content about music gear used by artists (instrument type, brand, model, etc.); from Deezer: songs' popularity rank, flag for explicit song lyrics, tempo, gain, song duration.

Merging was necessary when different properties of the same meaning coming from different data sources provided different, possibly complementary results (e.g. using *owl:sameAs* from DBpedia). The disambiguation properties of Discogs and the availability of multiple URIs from different data sources for the same song/artist/album also made it possible to detect gross errors. Conflict detectors were set up (e.g. for dates) and manual arbitration occasionally took place. We organized "WASABI Marathons" along the project lifetime where participants used the Wasabi Explorer [6] to identify errors. These marathons helped set up scripts to detect and fix errors ranging from spelling errors/variations (e.g. "Omega Man" by The Police is sometimes spelled "$\Omega$ Man"), to rules (e.g. several producers for the same song by the same artist generally indicate an anomaly). Table 2 shows the contribution of each data source in the final dataset e.g. we found in MusicBrainz 1,197,540 songs (57% of the 2,099,287 songs retrieved from LyricsWikia, our initial seed). The tools used to collect these metadata are available on the GitHub repository of the WASABI dataset.

### 3.2 Generating lyrics metadata

Lyrics encode an important part of the semantics of a song. We proposed natural language processing methods to extract relevant information, such as:

- *structural segmentation*: we trained a Convolutional Neural Network to predict segment borders in lyrics from self-similarity matrices (SSM) encoding their repetitive structure. Songs are therefore associated to labeled text segments corresponding to verse, chorus, intro, etc. [14]. We showed that combining text and audio modalities improves lyrics segmentation [15].
- *topics*: we built a topic model on the lyrics of our corpus using Latent Dirichlet Allocation (LDA). These topics can be visualized as word clouds of the most characteristic words per topic [13,10];
- *explicitness of the lyrical content*: we compared automated methods ranging from dictionary-based lookup to state-of-the-art deep neural networks to automatically detect explicit content in English lyrics [11];

- *salient passages of a song*: we introduced a method for extractive summarization of lyrics that relies on the intimate relationship between the audio and the lyrics (audio thumbnailing approach) [12];
- *emotions conveyed*: we trained an emotion regression model using BERT to classify emotions in lyrics based on the valence-arousal model [13].

Table 3 gives an overview of the annotations we published relating to the song lyrics. Some of those annotation layers are provided for all the 1.73M songs with lyrics included in the WASABI corpus, while some others apply to subsets of the corpus, due to various constraints of the applied methods [13].

**Table 3.** Song-wise annotations - ♣ indicates predictions of our models.

| Annotation | Labels | Description | Annotation | Labels | Description |
|---|---|---|---|---|---|
| Lyrics | 1.73M | segments of lines of text | Languages | 1.73M | 36 different ones |
| Genre | 1.06M | 528 different ones | Last FM id | 326k | UID |
| Structure | 1.73M | SSM $\in \mathbb{R}^{n \times n}$ (n: length) | Social tags | 276k | $\mathbb{S}$ = {rock, joyful, 90s, ...} |
| Emotion tags | 87k | $\mathbb{E} \subset \mathbb{S}$ = {joyful, tragic, ...} | Explicitness ♣ | 715k | True (52k), False (663k) |
| Explicitness | 455k | True (85k), False (370k) | Summary♣ | 50k | four lines of song text |
| Emotion | 16k | (valence, arousal) $\in \mathbb{R}^2$ | Emotion♣ | 1.73M | (valence, arousal) $\in \mathbb{R}^2$ |
| Topics♣ | 1.05M | Prob. distrib. $\in \mathbb{R}^{60}$ | | | |
| Total tracks | 2.10M | diverse metadata | | | |

The annotated corpus and the proposed methods are available on the project GitHub repository. As for structure segmentation, for each song text we make available an SSM based on a normalized character-based edit distance on two levels of granularity to enable other researchers to work with these structural representations: line-wise similarity and segment-wise similarity. As for lyrics summarization, the four-line summaries of 50k English lyrics is freely available within the WASABI Song Corpus, as well as the Python code of the applied summarization methods.[7] Concerning the explicitness of the lyrics content, we provide both the predicted labels in the WASABI Song Corpus (715k lyrics, 52k tagged as explicit) and the trained classifier to apply it to unseen text. As for emotion, the dataset integrates Deezer's valence-arousal annotations for 18,000 English tracks[8] [8], as well as the valence-arousal predictions for the 1.73M tracks with lyrics. We also make available the Last.fm social tags (276k) and emotion tags (87k) to allow researchers to build variants of emotion recognition models. Finally, we provide the topic distribution of our LDA topic model for each song and the trained topic model for future research.

### 3.3 Extracting chords through automatic audio content analysis

We enriched the dataset with automatic chord recognition [23] for reasons of consistency, formatting and coverage. Even though automatic transcriptions are

---

[7] https://github.com/TuringTrain/lyrics_thumbnailing

[8] https://github.com/deezer/deezer_mood_detection_dataset

not flawless, at least they are consistent and well-structured. Chord extraction can also be applied to each song for which audio is available, and therefore avoids the popularity bias that would follow from scraping crowd-sourced resources. In order to mitigate algorithmic imperfections, we used a chord recognition algorithm that additionally returns a song-wide measure of confidence in the quality of transcription [22]. Weighing by this measure makes dataset aggregations more reliable, although it obviously is of limited use when an individual song of interest has a particularly low confidence associated with it. A chord vocabulary consisting of sixty chords was imposed: all combinations of twelve possible root notes with five chord types (major, minor, dominant 7th, major 7th, minor 7th).

The Deezer song identifiers provided a link to audio recordings. The actual analysis required access to the raw, unprotected audio and was therefore run by Deezer on their servers. 1.2 million songs in the WASABI dataset have an associated Deezer identifier, so can potentially be enriched with a chord transcription. This process is still ongoing, currently 513K songs have been processed. The chord symbols (without timing due to copyright restrictions) and their confidence measures can be obtained along with the rest of the dataset through the REST API and SPARQL endpoint. Other musical properties based on automatic content analysis can be integrated as future work.

### 3.4   IRCAM tools for on-demand audio/MIR analysis

Timeside is an an open, scalable, audio processing framework in Python, enabling low and high level audio analysis, visualization, transcoding, streaming and labelling. Its API supports reproducible and extensible processing on large datasets of any audio or video format. For WASABI, some parts have been created or extended: a secured and documented API with JWT access capabilities, a Provider module to handle automatic extraction of YouTube's and Deezer's tracks or 30 seconds extracts, an SDK for the development of client applications and a new web front-end prototype. A hosted instance has been connected to the main WASABI API so that every available tracks can be dynamically processed and played back through the multi-track analyzer web player.

## 4   Formalizing, generating and publishing the RDF Knowledge Graph

The WASABI dataset essentially consists of two parts: the initial dataset produced over the last 3 years by integrating and processing multiple data sources, as explained in section 3, and the RDF dataset derived thereof, namely the *WASABI RDF Knowledge Graph* that we describe hereafter. The latter provides an RDF representation of songs, artists and albums, together with the information automatically extracted from lyrics and audio content.

### 4.1   The WASABI Ontology

The WASABI vocabulary is an OWL ontology to formalize the metadata. It primarily relies on the Music Ontology [24] that defines a rich vocabulary for describing and linking music information, and extends it with terms about some specific entities and properties. It also reuses terms from the Dublin Core, FOAF, Schema.org and the DBpedia ontologies, as well as the Audio Features Ontology[9] and the OMRAS2 Chord Ontology.[10]

The current version of the ontology comprises of 8 classes and 50 properties to describe songs, albums and artists. The number of properties is due to the quite specific features represented in the metadata, for instance multi-track files, audio gain, names and titles without accent, or lyrics-related features such as the detected language or explicitness. Furthermore, no less that 22 properties represent links to the web pages of social networks, either mainstream or specialized in the music domain. Whenever possible, we linked these properties with equivalent or related properties from other vocabularies.

The Music Ontology comes with three terms to represent music performers: *mo:SoloMusicArtist* and *mo:MusicGroup* that are both subsumed by class *mo:MusicArtist*. To distinguish between a music group, an orchestra and a choir, we defined the *wsb:Artist_Person* and *wsb:Artist_Group* classes, respectively equivalent to *mo:SoloMusicArtist* and *mo:MusicGroup*, and two subclasses of *mo:MusicArtist* namely *wsb:Orchestra* and *wsb:Choir*. *wsb:Song* is the class of musical tracks performed by artists. It's a subclass of *mo:Track*, itself a subclass of *mo:MusicalManifestation*. *wsb:Album* is the class of collections of one or more songs released together. It is a subclass of *mo:Record* which itself is a subclass of *mo:MusicalManifestation*.

The ontology namespace[11] (prefix *wsb:*) is also its URI. The ontology can be dereferenced with content negotiation, as well as all the terms of the ontology. It can be downloaded from the repository[12] where graphical visualizations are also available.

### 4.2   Representing songs, artists and albums in RDF

Beyond the terms of WASABI ontology, the resource descriptions use terms from multiple vocabularies. The namespaces and prefixes are given in Listing 1.1 and the diagram in Figure 2 is the representation of a song from the WASABI database. This song is linked to its album through the *schema:album* property, and both the album and song are linked to the artist using the *mo:performer* Music Ontology property. Only a small subset of songs' metadata is depicted here: title (*dcterms:title*), audio gain (*wsb:gain*), chords (*chord:chord*)[13] given

---

[9] Audio Features Ontology: `http://purl.org/ontology/af/`

[10] OMRAS2 Chord Ontology: `http://purl.org/ontology/chord/`

[11] `http://ns.inria.fr/wasabi/ontology/`

[12] `https://github.com/micbuffa/WasabiDataset/tree/master/ontology`

[13] Due to copyright concerns, the chords ordered sequence and timing were computed but are not provided.

```
@prefix af:      <http://purl.org/ontology/af/>.
@prefix chord:   <http://purl.org/ontology/chord/>.
@prefix dcterms: <http://purl.org/dc/terms/> .
@prefix foaf:    <http://xmlns.com/foaf/0.1/>.
@prefix mo:      <http://purl.org/ontology/mo/>.
@prefix rdf:     <http://www.w3.org/1999/02/22-rdf-syntax-ns#>.
@prefix rdfs:    <http://www.w3.org/2000/01/rdf-schema#>.
@prefix schema:  <http://schema.org/>.
@prefix wsb:     <http://ns.inria.fr/wasabi/ontology/>.
@prefix xsd:     <http://www.w3.org/2001/XMLSchema#>.
```

**Listing 1.1.** Namespaces used in the RDF representation of the entities.

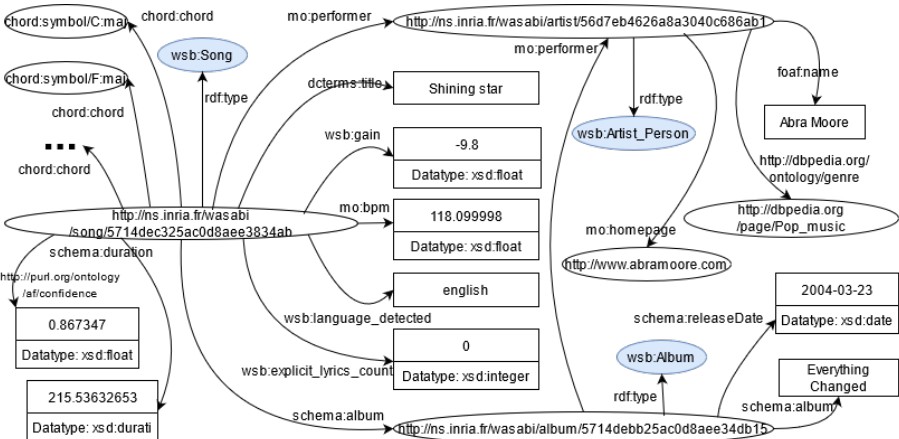

**Fig. 2.** RDF representation of a song, some properties and its artist and album.

by their URI in the OMRAS2 chord ontology, and number of explicit lyrics (*wsb:explicit_lyrics_count*).

## 4.3   The RDF Knowledge Graph Generation Pipeline

The dataset described in section 3 consists of a MongoDB database comprising three main collections: songs, artists and albums. The song collection provides not only the metadata, but also a representation of the chords extracted by the audio analysis, and the information extracted from the lyrics.

In a first stage, each JSON document of MongoDB was pre-processed so as to facilitate its translation to RDF, then translated to an RDF representation as described in section 4.2. The translation was carried out by Morph-xR2RML,[14] an implementation of the xR2RML mapping language [19] for MongoDB databases. All files involved in this pipeline are provided under the Apache License 2.0.

---

[14] https://github.com/frmichel/morph-xr2rml/

**Table 4.** Statistics of the WASABI dataset.

|  | No. entities | JSON data | No. RDF triples |
|---|---|---|---|
| Songs | 2.08 M | 8.8 GB | 49.9 M |
| Artists, groups, choirs and orchestras | 77 K | 378 MB | 2.76 M |
| Albums | 208 K | 424 MB | 3.29 M |
| **Total** | 2.38 M | 9.57 GB | 55.5 M |

**Table 5.** Selected statistics on typical properties and classes.

| Property URI | nb of instances | comment |
|---|---|---|
| http://purl.org/ontology/chord/chord | 7595765 | chords of a song |
| http://purl.org/dc/terms/title | 2308022 | song or album title |
| http://schema.org/album | 2099283 | song-to-album relation |
| http://purl.org/ontology/mo/performer | 1953416 | performing artist |
| http://www.w3.org/2002/07/owl#sameAs | 204771 | DBpedia/Wikidata links |
| http://purl.org/ontology/mo/genre | 86190 | musical genre |
| http://purl.org/ontology/mo/producer | 75703 | song/album producer |
| http://schema.org/members | 74907 | group members |
| http://dbpedia.org/ontology/genre | 52047 | DBpedia musical genre |

| Class URI | nb of instances | |
|---|---|---|
| http://ns.inria.fr/wasabi/ontology/Song | 2099287 | |
| http://ns.inria.fr/wasabi/ontology/Album | 208743 | |
| http://ns.inria.fr/wasabi/ontology/Artist_Group | 29806 | group or band |
| http://ns.inria.fr/wasabi/ontology/Artist_Person | 24264 | single artist |
| http://purl.org/ontology/mo/MusicArtist | 23323 | |
| http://ns.inria.fr/wasabi/ontology/Classic_Song | 10864 | classic of pop/rock music |
| http://ns.inria.fr/wasabi/ontology/Choir | 44 | |
| http://ns.inria.fr/wasabi/ontology/Orchestra | 30 | |

### 4.4   Publishing and Querying the WASABI RDF Knowledge Graph

Table 4 synthesizes the amount of data processed to produce the WASABI RDF Knowledge Graph, and reports the number of triples produced. Table 5 reports some statistics about the instances.

**Dataset Description and Accessibility.** In line with data publication best practices [9], the WASABI RDF Knowledge Graph comes with rich metadata regarding licensing, authorship and provenance information, linksets, vocabularies and access information. These can be visauliazed by looking up the dataset URI[15]. The dataset is available as a DOI-identified downloadable RDF dump and a public SPARQL endpoint (see Table 6). All URIs can be dereferenced with content negotiation. Further information (modeling, named graphs, third-party vocabularies) are documented in the GitHub repository.

**Dataset Licensing.** Like the rest of the WASABI dataset, the WASABI RDF Knowledge Graph is published under the Creative Commons Attribution-NonCommercial-ShareAlike 4.0 International License. Copyrighted data such as

---

[15] WASABI RDF dataset URI: http://ns.inria.fr/wasabi/wasabi-1-0

**Table 6.** Dataset accessibility.

| RDF dump | `https://doi.org/10.5281/zenodo.4312641` |
|---|---|
| Public SPARQL endpoint | `http://wasabi.inria.fr/sparql` |
| Documentation | `https://github.com/micbuffa/WasabiDataset` |
| Ontology namespace | `http://ns.inria.fr/wasabi/ontology/` |
| Data namespace | `http://ns.inria.fr/wasabi/` |
| Dataset URI | `http://ns.inria.fr/wasabi/wasabi-1-0` |

the full text content of the song lyrics and audio are not included but URLs of original source material are given.

**Sustainability Plan.** We plan several research lines that will exploit and extend the dataset and improve its resilience. We intend to add more audio-related computed metadata from our collaborators at IRCAM and QMUL e.g. links from the song ids to the TimeSide API will ensure that the audio based analyses can be provided or re-triggered even if new songs are added or of the audio data change or vary from the external providers. We also deployed a SPARQL endpoint that benefits from a high-availability infrastructure and 24/7 support.

## 5    Visualization and Current Usage of the Dataset

The size and the complexity of the dataset require appropriate tools to allow users to explore and navigate through it. This multidimensional dataset contains a large variety of multimedia attributes (lyrics, sounds, chords, musical instruments, etc.) that are interlinked, thus featuring a large and rich knowledge graph. To assist users, we investigate various visualization techniques.

Our goal is to help users explore the dataset by providing answers to common visualization questions such as to get an overview of itemsets fitting some user-defined criteria, exploring details of particular itemsets, identify relationships (such as patterns, trends, and clusters) between itemsets, etc. These common tasks are defined in the information-seeking mantra introduced by Schneidermann [25] which guides the design of all visualization tools. We designed and implemented a large set of visualization techniques using the D3.js library and made these techniques available to the user in a gallery. Figure 3 illustrates some of the visualization techniques currently available and all are interactive so that user can select an itemset and apply zoom and filtering to explore the dataset.

The creation process for the gallery of information visualization techniques is rather opportunistic and incremental. It allows us to explore different alternatives for showing information to the users but also to combine different attributes. The visualization is driven by the type and inner structure of the data that results from the queries embedded into the visualization tools. Whilst visualization tools created this way are not generic, they remove part of the inner complexity of creating (SPARQL) queries, making the tools easier to use and particularly suitable to communicating results to a broad audience.

It is worthy of notice that some of the visualization techniques includes multimedia content, for example the Figure 4 includes images that refer to cover of

albums of an artist. So far, beyond the interactive graphics, only text and images are used in multimedia visualization, we are working to enrich the information visualization gallery with techniques that include audio contents.

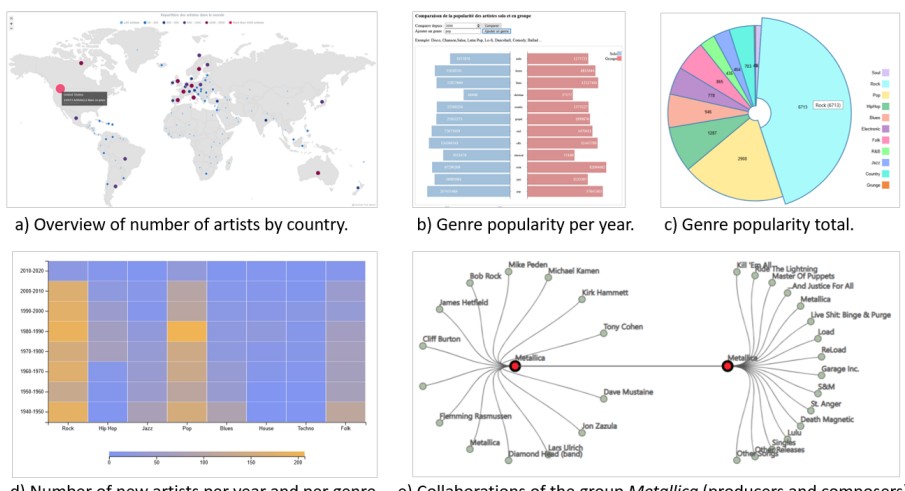

a) Overview of number of artists by country.      b) Genre popularity per year.      c) Genre popularity total.

d) Number of new artists per year and per genre.   e) Collaborations of the group *Metallica* (producers and composers)

**Fig. 3.** Selected visualization techniques from the WASABI gallery: a) *map view* showing artists per country; b) *barplot* showing a comparison of the popularity of two genres along years; c) *pie chart* showing total genre popularity; d) *heatmap* showing the number of new artists of a genre per year; e) *bifocaltree* showing collaborators of Metallica.

## 6   Evaluating the quality of the dataset, future updates

Assessing and ensuring the quality of a large dataset built by aggregating multiple data sources is challenging and continuous process. It is common that metadata coming from various sources be erroneous or conflicting. The multiple hackathons helped in spotting many errors, conflicts and recurring problems, and we wrote a set of scripts (available in the Github repository) to fix some of them. The dataset is meant to be maintained over at least the next three years as we have new ongoing projects that will exploit and extend it. More metadata will be added (in particular from MIR audio analysis of songs, linking songs to existing midi transcriptions or online scores), and we are developing new metadata quality assessment tools (based on visualisations and inferences rules) that shall allow the community to better detect and report erroneous, conflicting or missing metadata.

On the other hand, the quality of extracted metadata about lyrics have been validated using different methods described in the papers cited in section 3.2.

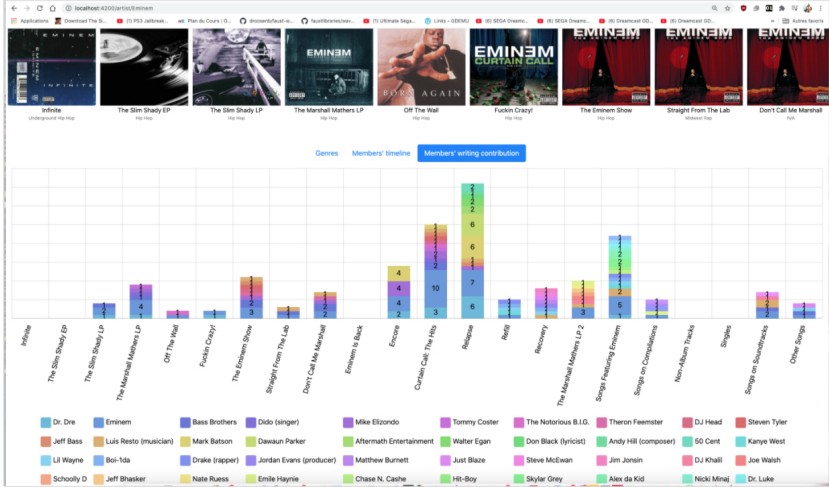

**Fig. 4.** Example of information visualization including multimedia contents: Eminem collaborations (co-writing of songs) by albums (see cover of albums).

## 7    Potential Impact and Reusability

To the best of our knowledge, the WASABI dataset is the first one integrating cultural and MIR data at this scale into a single, coherent knowledge graph that makes it possible to initiate new research. The wide range of metadata resulting from the analysis of song lyrics is one of the remarkable points. Many recommendation systems are based on cultural information and user profiles, sometimes taking into account data from the audio analysis, but few rely on the content of the lyrics or their analysis. Furthermore, mixing the emotions extracted from the textual analysis with results on the analysis of emotions extracted from the audio from other datasets (e.g. AcousticBrainz) provides new prospects for recommendation systems [20]. In addition to being interoperable with central knowledge graphs used within the Semantic Web community, the visualizations show the potential of these technologies in other fields. The availability of this rich resource can also attract researchers from the NLP community.[16]

**Interest of communities in using the Dataset and Services.** The openness of the data and code allow contributors to advance the current state of knowledge on the popular music domain. We initiated collaborations with other groups and, in particular, researchers from the FAST project already used and contributed to the dataset. IRCAM researchers also started cross-domain analyses on songs (i.e structure and emotion detection using both audio and text[18]). Collaboration with IRCAM will continue as the integration of additional audio data is a priority for us in the coming months and will trigger updates.

**Application scenarios, targeted users, and typical queries.** Following a user-oriented requirement analysis, we designed a set of motivating scenarios:

---

[16] 1st Workshop on NLP for Music and Audio in 2020 https://sites.google.com/view/nlp4musa

*Scenario 1: research of artists and songs related to current events*, set with the help of archivists from Radio-France. E.g. during the "yellow jackets" protest in France, animators of music radio programs repeatedly requested songs about protests, rebellion, anti-government movements, revolution.

*Scenario 2: analysis of a particular artist*, set with the help of musicologists. To look for an artist's influences, collaborations and themes, and study variations in the compositions (complexity of the songs, recording locations).

*Scenario 3: disambiguation of homonyms (artists with the same name) and duplicates (single artist with multiple names or various name spellings)*, set with the help of Deezer. The goal is to highlight suspicious artist profiles and the sources involved (e.g. homonyms may have different record labels, languages etc.), to display, group and prioritize alerts (e.g. using artists' popularity).

*Scenario 4: search for songs of a particular style, in a given key, or containing specific chords, with lyrics with given topics*, set with the help of music schools and professional composers. E.g.: look for a blues in E with a tempo of 120 bpm, then get similar songs but with chords outside the key; show artists who wrote a type of songs, sorted by popularity; search for songs with given themes and emotions in the lyrics or with certain types of orchestrations (e.g. guitar and clarinet). Like in scenario 1, these queries involve searches through the dataset, yet they are meant for very different users with different needs that require specific user interfaces: journalists vs. music schools and composers.

Whilst some questions might be answered by showing the correlation between components (e.g., types of collaborations between artists), others might require reasoning (e.g., compute the possible keys from the list of chords), and mix cultural, audio or lyrics related content (e.g., orchestration mixes audio and metadata collected about artists/members' instruments). Answering these complex queries might also require an exploration of the WASABI corpus, and for that we offer a variety of analysis, exploration and visualization tools [6].

## 8   Conclusion and Future Work

In this paper, we described the data and software resources provided by the WASABI project to make it easier for researchers to access, query and make sense of a large corpus of commercial music information. Applications meant for different types of users (composers, music schools, archivists, streaming services) can be built upon this corpus, such as the WASABI explorer that includes a chord search engine and an augmented audio player[6].

We generated and published an RDF knowledge graph providing a rich set of metadata about popular commercial songs from the last six decades. These metadata cover cultural aspects of songs/albums/artists, lyrics content as well as audio features extracted using Music Information Retrieval techniques. The RDF representation currently uses some free text values coming from the original sources. To be more in line with Linked Data practices, in the future we intend to improve this by reusing, extending or defining thesauruses with respect to musical genres, music instrument types (currently 220 distinct values

among which 16 different types of guitar) and equipment types (e.g. microphone, amplifier). We also published the pipeline we set up to generate this knowledge graph, in order to *(1)* continue enriching it and *(2)* spur and facilitate reuse and adaptation of both the dataset and the pipeline.

It is important to note that during the project we had access to copyrighted content (song lyrics, audio files) that we are not allowed to include in the published dataset. Nevertheless, a lot of work has been done on the analysis of song lyrics and the results are available on the GitHub of the project (metadata, ML templates, Python scripts and Jupyter notebooks). It is still possible for researchers to get the lyrics (e.g. via the commercial MusixMatch API[17]) or the audio (via Deezer's public API which offers 30-second clips, or via the YouTube API for example) to reproduce our results. Some computed metadata related to the synchronized chord sequences (used by the augmented audio player in the WASABI online explorer) were published only partially to avoid copyright infringement (as they are too close to music scores). Nevertheless, in collaboration with Deezer and IRCAM, we plan to carry out further audio analyses, mainly for scenarios of interest to music schools, musicologists, archivists and broadcast services.

Results from the lyric processing are provided in the original dataset in different forms (json, csv, etc.). At the time of writing, only the explicitness metadata are included in the RDF knowledge graph, and we are working on a future update that will include other metadata. However, all the components are connected through the WASABI ids, shared by all versions of the dataset.

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
