# OpenReview forum: "The WASABI dataset: cultural, lyrics and audio analysis metadata about 2 million popular commercially released songs"
_eswc-conferences.org/ESWC/2021/Conference/Resources_Track — ESWC 2021 Resources_

### Official Review · AnonReviewer1 · 2021-01-05
**A knowledge graph of popular music linking lyrics, topics, and audio**

**Rating:** 1
**Confidence:** 5

**Review:**

I thank the authors for their comprehensive answer to my feedback in the rebuttal. I don't have additional concerns except noticing how numerous changes to the draft should be performed in case of acceptance. I am still inclined to accept the paper for publication in the conference.
***

WASABI is a large dataset of commercial popular music built combining information from several existing open datasets and including lyrics, chord information, audio summaries, as well as topical and cultural annotations. The dataset is partly the result of an aggregation of existing sources. It also publishes the output of several NLP and MIR pipelines, applying state of the art research for lyrics segmentation, emotion classification, and explicit content detection, among others.
The project developed several visualisations. However, how domain experts such as music historians and musicologists interested in popular music could access and use such a resource? Besides, authors should discuss WASABI in relation to other semantic web projects on theme, such as the DOREMUS [1], MELD [2], MIDI LD [3],  and the LED project [4].

[1] Lisena, Pasquale, Manel Achichi, Pierre Choffé, Cécile Cecconi, Konstantin Todorov, Bernard Jacquemin, and Raphaël Troncy. "Improving (re-) usability of musical datasets: An overview of the DOREMUS project." Bibliothek Forschung und Praxis 42, no. 2 (2018): 194-205.

[2] Page, Kevin, David Lewis, and David Weigl. "MELD: a linked data framework for multimedia access to music digital libraries." In 2019 ACM/IEEE Joint Conference on Digital Libraries (JCDL), pp. 434-435. IEEE, 2019.

[3] Meroño-Peñuela, Albert, Rinke Hoekstra, Aldo Gangemi, Peter Bloem, Reinier de Valk, Bas Stringer, Berit Janssen et al. "The MIDI linked data cloud." In International Semantic Web Conference, pp. 156-164. Springer, Cham, 2017.

[4] Adamou, Alessandro, Simon Brown, Helen Barlow, Carlo Allocca, and Mathieu d’Aquin. "Crowdsourcing Linked Data on listening experiences through reuse and enhancement of library data." International Journal on Digital Libraries 20, no. 1 (2019): 61-79.



**Anonymity:**

No, I would like my review to be deanonymized.

**Strong Points:**

The resource is the main output of the WASABI project (http://wasabihome.i3s.unice.fr/), which combines MIR, NLP, and Semantic Technologies to deliver a large scale knowledge graph on popular music.
The graph includes an impressive amount of links to external sources (example: https://wasabi.i3s.unice.fr/#/search/artist/Frank%20Zappa).
The data is available and can be queried openly (http://wasabi.inria.fr/describe/?url=http://ns.inria.fr/wasabi/song/5714ded025ac0d8aee423ac7).
The resource is a fascinating example of the heterogeneity of domain-specific knowledge graphs, which opens several exciting questions, especially on the side of information visualisation and exploration.



**Subreviewer:**

I submitted this review.

**Weak Points:**

Although the project reused ontologies from the SW community, mainly the Music Ontology, the paper should have better placed the dataset in relation to other efforts on semantics and music(ology). These include the MIDI Linked Data, a project that generated a huge corpus of MIDI files represented as linked data. Such symbolic music information is complementary to the type of data included in WASABI. Alignments could be exploited for checking the quality of the detected chords, for example. The work of the DOREMUS project is also a relevant antecedent in the music domain and the Listening Experience Database. It would be interesting to see these datasets fully interlinked sometime in the future.

The application scenarios described are interesting, but it looks like the target users of such resource should be musicologists of popular music or scholars studying a cultural history of popular music. What type of tools/methods are needed to approach this audience?

The search/exploration interface could be improved further, with many broken images on my chrome browser (https://wasabi.i3s.unice.fr/#/search/infos/more/bob%20dylaN).

---

> ### Author Rebuttal · Authors · 2021-01-29
>
> We thank the reviewer for his/her valuable remarks. Below we answer the different concerns raised, and we briefly explain the changes we will make to the camera ready version of the paper, if accepted.
>
> > how domain experts such as music historians and musicologists interested in popular music could access and use such a resource?
>
> We will clarify this in the final paper.
> A professional composer, a music school (several music teachers)  and a musicologist collaborated with the project, in addition to IRCAM collaborators who are also music experts. Aside the dataset, interactive applications have been or are being developed. We have an ongoing PhD about music data visualization (targeting musicologists) that will exploit the dataset.
>
> During the project, IRCAM developed the Timeside service, an online tool linked to the dataset and meant for musicologists. It includes an augmented audio player with annotation tools for musicologists, audio analysis on demand, etc.  A “search by chords” tool is also already integrated in the WASABI online explorer that was one of the requests by the composers who participated in the user need campaign.
>
> Furthermore, we are developing new tools that will exploit the semantic aspects of the metadata (i.e the MG-explorer library we’re using for visualizations). Some ongoing works are also adding inference rules to the existing ontology for special use cases (i.e visualizing the influences of one artist or producer on another one, etc.) And we will involve these human experts in the new evaluation campaign to come.
>
> > authors should discuss WASABI in relation to other semantic web projects on theme, such as the DOREMUS [1], MELD [2], MIDI LD [3], and the LED project [4].
> > Although the project reused ontologies from the SW community, mainly the Music Ontology, the paper should have better placed the dataset in relation to other efforts on semantics and music(ology) (...).
> > Alignments could be exploited for checking the quality of the detected chords, for example. The work of the DOREMUS project is also a relevant antecedent in the music domain and the Listening Experience Database. It would be interesting to see these datasets fully interlinked sometime in the future.
>
> There are indeed a number of reference works for which we will add a comparison in the related works section.
>
> Some goals of the DOREMUS project overlap with the WASABI yet in a rather different context. It focuses on classical and traditional music (popular songs in WASABI), and does not tackle audio and lyrics processing. However, interesting methods have been used in DOREMUS, e.g. the integration of MIDI resources, recommendation, automatic playlists generation, where a joint exploitation of both datasets could be fruitful.
>
> The WASABI ontology extends the Music Ontology (MO), yet the Performed Music Ontology (part of LD4L) or DOREMUS ontology may be considered if future works need to model more accurately the differences between works, performances, expressions etc.
>
> The Listening Experience Database collects people’s music listening experiences as they are reported in documents like diaries or letters. It mostly relates to legacy music that has little overlap with WASABI.
>
> The MELD framework supports the publication of musicology articles with multi-modal user interfaces that connect different forms of digital resources. Some development could be undertaken to allow musicologists publish articles that would leverage musical data from the WASABI RDF knowledge graph.
>
> The MIDI Linked Data project publishes a large set of MIDI files in RDF. Linked to DBpedia and relying on the MO, it could complement WASABI to jointly exploit MIDI files and audio and text analyses. Interestingly, some MIDI content was used in WASABI during the evaluation of the chords extraction.
>
>
> > The application scenarios described are interesting, but it looks like the target users of such resource should be musicologists of popular music or scholars studying a cultural history of popular music. What type of tools/methods are needed to approach this audience?
>
> We started a PhD about music data visualization, that will be suited for musicologists. The ongoing integration of IRCAM Timeside service will allow musicologists to perform audio analysis on demand and annotate songs interactively. Music visualization is not only graphics but also audio exploration. We have already developed many WebAudio tools related to the dataset, for music schools and for musicologists. See our publications at the WebAudio conference, for example:
> Michel Buffa, Jerome Lebrun, Johan Pauwels, Guillaume Pellerin. A 2 Million Commercial Song Interactive Navigator. WAC 2019 - 5th WebAudio Conference 2019, Dec 2019, Trondheim, Norway. ⟨hal-02366730⟩
>
> > The search/exploration interface could be improved further, with many broken images on my chrome browser
>
> This was due to a problem with an https certificate. Sorry about that, it should be fixed now.

---

> > ### Comment · AnonReviewer1 · 2021-02-02
> > **Exhaustive feedback to the concerns raised in the review**
> >
> > I thank the authors for their comprehensive answer to my feedback. I don't have additional concerns except noticing how numerous changes to the draft should be performed in case of acceptance. I am still inclined to accept the paper for publication in the conference.

---

### Official Review · AnonReviewer2 · 2021-01-12
**a notable amount of work for producing a valuable resource**

**Rating:** 2
**Confidence:** 4

**Review:**

The authors introduce the WASABI dataset, a very large dataset consisting of metadata from different open sources, originated from both explicit information and data obtained through processing of unstructured information. The dataset includes music information such as lyrics and chords (even though limited in their representation, with no time information nor sequencing) in explicit and derived form (e.g. topics, emotions) as well as other extra-musical content such as different kind of social connections among artists.

The resource offers both a RDF dataset (limited, for now, to the explicit metadata) and a database (powered by MongoDB) backing a large set of services, including different visualizations for exploring and analyzing the data.

I am impressed by the large amount of work that has been done in putting several, different, aspects of the music material together. While it is true that the Semantic Web does not have to be populated of few, monolithic, resources containing everything, and that linking different datasets is at the base of providing this mashup on a distributed scale, it is also true that not all of the original sources involved in this operation had published Linked Datasets on the Web, thus big RDF mashups are always welcome while the Web is still far from being overpopulated by datasets. About that, by resolving resources on the Web, I found owl:sameAs for artists (to wikidata and dbpedia) but none for songs. Was there no other RDF dataset among those adopted to import the songs?

A part that seems missing from the article is related to the evaluation of the quality of the extracted content. I understand that this is a choral paper describing WASABI as a whole and it cannot go into the details of every aspect (and it has been appropriate, in this sense, to report references to other publications that verticalize on specific topics, or to the GitHub site for further details), however I am missing – as a dedicated section – anything related to the evaluation of the content – which is a large part of the dataset – that has been obtained through processing techniques of various sorts. I see though that this evaluation is available in some – not all – of the referenced papers, yet it would be good to have at least a dedicated section providing a bird’s view over the quality of the generated contents. The only mention of this concept that I found is about some metadata getting a reliability value, so that whoever uses it knows, wrt the implicit reliability scale defined and applied by the method itself, how much the information is reliable. No indication is provided of how the application of the indicated techniques to the said data has produced meaningful results, subjected to the evaluation of human experts. I mean, while I have no reason to object the quality of the processed content, there is little in the paper that guarantees that this is not a huge amount of garbage added to mere explicit metadata.

A tiny detail that could be better clarified: in page 4 variations are addressed as an error. Indeed technically, if speaking about the title, it is (there is only one official title with one spelling only) yet it could be interesting to address these variations (e.g. the omega from the song of The Police) and have them maintained and properly represented.

The Sustainability Plan described in pag. 10 seems to describe what will continue to happen during the project (i.e. data addenda from partners; quality of service) but doesn’t make a precise statement about life of the resource after the project. Will INRIA continue to curate (or at least maintain) the resource online or will some other partner take the lead? There is only a mention of past-project collaboration with IRCAM on page 14.

Despite my point on evaluation, I would gladly see this paper accepted.

TECHNICAL NOTES:

* I tried to import the ontology. I tried both on VocBench and Protégé: both succeeded in doing it and both had issues with many failed imports for ontologies, imported by the Music Ontology, that are not resolvable. VocBench did it quickly, reporting the failed imports on the import tree, so that it is possible to repair them (e.g. by loading the file from the filesystem). Protege took a long time, probably making more attempts for each import, and then interactively asking me for further attempts or simply if going ahead without importing the problematic ontologies. Surpassed this step, I noticed however, to my surprise, that there was no content from the main ontology (the WASABI one). I thus made a SPARQL query on the graph of the main ontology (i.e. excluding the imports) and found only triples (19 of them) with the ontology as the subject. I then realized what happened: the ontology has been loaded on the WASABI SPARQL endpoint and not made available as a full file. This way, the resolution of its URI returns the description of the sole entity describing the ontology itself, not the whole ontology. Unless I am unaware of some other way (recommendation, best practice etc..) to deal with it (e.g. some metadata telling a tool performing http resolution of the ontology term to download also the other terms) and which should have been supported by the above tools, I guess it is better to resolve the URI of the ontology on the whole ontology file

* I resolved the sample resource presented in figure 2: http://ns.inria.fr/wasabi/song/5714dec325ac0d8aee3834ab. I found no void:inDataset triple. What has been the policy adopted for letting machines automatically discover the VoID file?

MINOR REMARKS and TYPOs:

* the table in pag.1 reports “English songs”. I suppose it’s meant to say songs with lyrics in the English language but I think that way reads songs produced/authored in England, as one of the four countries of UK

* in page 5, expand the acronym BERT (Bidirectionnal Encoder Representations for Transformers) and provide a link/reference to it

* last paragraph of conclusions: explicitness metadata --> explicit metadata

Post-Rebuttal Comments
=========================

I thank the reviewers for addressing my concerns. Indeed, the absence of evaluation of the quality of extracted/processed data is a severe flaw, yet I keep my evaluation due to the relevance of the resource (including many aspects that are not subject to processing and are thus reliable). Concerning the issue with the ontology file, the authors say “tools such as the ones you mention” but mines were just examples. Like I said, the problem is more general and not tool-specific: URIs of ontologies resolve into the entire description of the ontology…unless I’m missing something.


**Anonymity:**

No, I would like my review to be deanonymized.

**Strong Points:**

* A huge amount of work producing a dataset large both in extensions (2 million songs, 200K albums, 77k artists) and details (lot of metadata for each element). The comparison with the state of the art suggests that this is indeed a valuable resource

* An omni-comprehensive resource including an RDF dataset, a set of advanced services and visualization functionalities.


**Subreviewer:**

I submitted this review.

**Weak Points:**

Missing, if not evaluation, at least some considerations/indications on the quality of the processed content

---

> ### Author Rebuttal · Authors · 2021-01-29
>
> We thank the reviewer for his/her valuable remarks. Below we answer the different concerns raised, and we briefly explain the changes we will make to the camera ready version of the paper, if accepted.
>
> >I found owl:sameAs for artists (to wikidata and dbpedia) but none for songs. Was there no other RDF dataset among those adopted to import the songs?
>
> This is indeed an issue that we noticed while translating the WASABI database to RDF: DBpedia and Wikidata ids are provided for the artists but not for songs for which they were probably left out unintentionally. In the new phase of the project that has recently started we intend to fix this issue.
>
> >A part that seems missing from the article is related to the evaluation of the quality of the extracted content. (...) I am missing – as a dedicated section – anything related to the evaluation of the content (...) that has been obtained through processing techniques of various sorts.
>
> > No indication is provided of how the application of the indicated techniques to the said data has produced meaningful results, subjected to the evaluation of human experts. (...)
>
> Indeed, this was not clear in the submitted paper, as several reviewers noticed. We will add a section about it.
>
> Some metadata coming from external data sources could be erroneous or conflicting. The "marathons" help in spotting many errors, conflicts and recurring problems, and we wrote a set of scripts (available in the git repository) to address some of them. However, this work is not finished and we’re regularly improving the quality of the dataset, when new problems are encountered. We are currently developing visualization tools and inference rules that can be used by the community to report problems (in a crowd-sourcing manner) and better detect conflicting metadata (internally or with other data sources).
>
>
> > it would be good to have at least a dedicated section providing a bird’s view over the quality of the generated contents.
>
> Indeed, we will add a new section in the final paper about evaluation, that will explain how music human experts have been involved and what methodology we used.
>
>
> > The Sustainability Plan described (...) doesn’t make a precise statement about life of the resource after the project. Will INRIA continue to curate (or at least maintain) the resource online or will some other partner take the lead?
>
> The dataset is meant to be maintained over at least the next three years as we have new project that will exploit it. More metadata will be added (in particular from audio analysis of songs), and a PhD thesis recently started on music data visualization on this dataset, which will address the problem of bad or missing metadata identification.
>
>
> > the ontology has been loaded on the WASABI SPARQL endpoint and not made available as a full file. (...)
> > I guess it is better to resolve the URI of the ontology on the whole ontology file
>
> Thank you for raising this point. Like any other URI in the dataset, the ontology's URI is dereferenced directly by our Virtuoso endpoint. While this is consistent, this proves to be a problem with tools such as the ones you mention. Still, it is possible to get more by querying all triples related to the ontology, this would return also the classes and properties that all have a triple "<x> rdfs:isDefinedBy <ontology>".
>
> We'll look more deeply into this recurring issue, to decide wether we should resolve the URI to the Turtle file, or alternatively create a new URI with ".ttl" appended at the end, to return specifically the Turtle file.

---

> > ### Comment · AnonReviewer2 · 2021-02-01
> > **thanks for the clarifications**
> >
> > I thank the reviewers for addressing my concerns. Indeed, the absence of evaluation of the quality of extracted/processed data is a severe flaw, yet I keep my evaluation due to the relevance of the resource (including many aspects that are not subject to processing and are thus reliable).
> > Concerning the issue with the ontology file, the authors say “tools such as the ones you mention” but mines were just examples. Like I said, the problem is more general and not tool-specific: URIs of ontologies resolve into the entire description of the ontology…unless I’m missing something.

---

### Official Review · AnonReviewer4 · 2021-01-14
**Rich song dataset**

**Rating:** 2
**Confidence:** 4

**Review:**

I have read the authors' rebuttal which clarifies some things but does not affect my review.
-----
The paper presents the WASABI dataset for enriched lyrics and audio metadata for 2 million songs. The paper is quite well written and there is a lot of potential use for such data. I do have some questions regarding the enrichments and some of the descriptions:

- What is meant by 'cultural' data on songs? The paper mentions this several times, but I could not find an explanation of what this entails exactly.
- Various analyses and enrichments were performed on the data, but the paper does not provide any performance indicators for this. I can imagine that the chord identification in songs with for example weird guitar tunings might be less reliable than on more standard tunings, and in particular for the NLP (which I know more about) emotion tagging and topic modelling do not always perform perfectly. How does WASABI deal with this? I see in the RDF representation in Figure 2 that there are some confidence measures, but it would be good to know more about this.
- Regarding the chords representation, in Figure 2, there are two chords associated with the example song: Cmaj and Fmaj, is there any sequential information as this is not apparent to me from the model. Also, there are many online scores/tabs/chord sheets available, would it be possible to link these?
- How would musicologists look for hidden relationships between artists (e.g. influences, indirect collaborations) to support a claim using WASABI (this is given as an example in Section 1. This is touched upon in Scenario 2 in section 6 but some more detail would be would be instructive to an audience wanting to use this resource.

Textual comments:
Section 1
- composers also collaborated to this project -> composers also contributed to this project (even better would be to explain what their role is)
- QMUL is only explained in Section 2.

Section 2
- organology^11. -> organology.^11 (the Oxford style guide recommends footnote markers to be placed outside punctuation, please also check other footnote markers)
- The last paragraph of section 2 (The Million Song Dataset ... of metadata) is a bit difficult to follow, perhaps it would help to split this up into two paragraphs, one specifically about other work, and the final one about how WASABI is different?

Section 3
- Table 1: provide column names to all columns (in particular the last column is a bit puzzling)
- [under submission] -> it would have been nice if this had been made available to the reviewers
- salient passages of a song -> it would have been helpful to me if there had been a bit more explanation here
- Table 2: there is a lot of interesting stuff here and it would have been nice if this had been described in more detail. For example which languages are in the 36 that are counted. Are the text enrichments only for English or also for the other languages?

Section 4
- please fix the text running into the margin at the bottom of page 8

Section 6
- inter-operable -> interoperable

Section 7
- Conclusion and Future Works -> Conclusion and Future Work

**Anonymity:**

No, I would like my review to be deanonymized.

**Strong Points:**

- rich resource
- well-written paper


**Subreviewer:**

I submitted this review.

**Weak Points:**

- evaluation missing
- text could be clarified in some cases (in particular with what is meant by cultural data, which is quite an important concept)
- use cases could be more detailed

---

> ### Author Rebuttal · Authors · 2021-01-29
>
> We thank the reviewer for his/her valuable remarks. Below we answer the different concerns raised, and we briefly explain the changes we will make to the camera ready version of the paper, if accepted.
>
> > evaluation missing
>
> You are right. Indeed, quality evaluation needs to be improved, this is a continuous process. Some metadata coming from external data sources could be erroneous or conflicting. For instance, some Wikipedia pages for a song describe the song itself, but sometimes also many cover versions. In such cases, the infobox (used by DBPedia extractors) could contain metadata for ALL cover versions at the same time.
>
> The hackathons help in spotting many errors, conflicts and recurring problems, and we wrote a set of scripts (available in the git repository) to address some of them.
> The dataset is meant to be maintained over at least the next three years as we have new project that will exploit it: more metadata will be added (in particular from audio analysis of songs). Furthermore, we are currently developing visualization tools and inference rules that can be used by the community to report problems (in a crowdsourcing maner), and to better detect conflicting metadata (internally or with other data sources). We will more clearly describe this continuous quality process in the paper.
>
> > What is meant by 'cultural' data on songs? The paper mentions this several times, but I could not find an explanation of what this entails exactly.
>
> In the context of this paper, cultural data is used in the sense of the artistic culture and all the music culture data available around the piece itself such as songs/albums/artists. We will include an example of a piece of data to clarify this at the beginning of the article.
>
> > use cases could be more detailed
>
> Indeed we had to summarize the use-cases so as to comply with the size constraint: we touched upon them in the introduction and got into just a few more details in section 6, describing several use cases: music streaming services, journalists or archivists, musicologists. If space permits, we will expand one of them.
>
> > and in particular for the NLP (which I know more about) emotion tagging and topic modelling do not always perform perfectly. How does WASABI deal with this? I see in the RDF representation in Figure 2 that there are some confidence measures, but it would be good to know more about this.
>
> Concerning the emotion tagging: we aligned the valence-arousal annotations for the 18,000 English tracks released by Deezer (derived from the method of Delbouys et al., 2018 arXiv:1809.07276) to our songs. Based on their annotations, we train an emotion regression model using BERT, with an evaluated 0.44/0.43 Pearson correlation/Spearman correlation for valence and 0.33/0.31 for arousal on the test set. As for topic modeling, we trained a topic model of 60 topics on the unique English lyrics in WASABI (1.05M). We have then manually labelled a number of more recognizable topics, and have illustrated these topics with word clouds of the most characteristic words per topic. More details on those tasks and their evaluation  can be found at https://hal.inria.fr/tel-02587910/file/mfell_thesis_song_lyrics_tel.pdf
>
> > Regarding the chords represe ntation, in Figure 2, there are two chords associated with the example song: Cmaj and Fmaj, is there any sequential information as this is not apparent to me from the model. Also, there are many online scores/tabs/chord sheets available, would it be possible to link these?
>
> The chords in the RDF representation are indeed not ordered. This restriction comes from Deezer's legal department who was concerned that a full, ordered list of chords would be too close to the music score that is protected by a copyright. Thus, although we can provide tools exploiting the ordered chords, we cannot redistribute it. We hope this is something that shall be rediscussed in the future.

---

### Official Review · AnonReviewer5 · 2021-01-16
**Relevant resource for the music domain with some unclear aspects**

**Rating:** 1
**Confidence:** 4

**Review:**

Post rebuttal comment:

I would like to thank the authors for their explanations.
I find the resource mature and relevant.
Nevertheless, I highly recommend that the authors elaborate more in the paper about:

1) the quality of the dataset?
2) how the dataset relates to other related works/datasets?

My comments from the initial review are provided below.
Thank you.

--


The paper describes a resource, i.e. dataset, containing music related information.
The paper is well written, well structured and easy to follow.
The work presented is of high quality, however there are two aspects which are not properly addressed:
1) It is unclear what is the quality of the published data
2) It is not clear to what extend the resource is complementary/differs w.r.t. the other related works/datasets.

Another minnor comment - Sec 4. It would be nice to add visualisation (figure) of the ontology in the paper.

No language or grammar issues have been spotted.

**Anonymity:**

No, I would like my review to be deanonymized.

**Strong Points:**

- The paper reads well and it is in general easy to follow.
- The metadata processing pipeline is well developed.
- The dataset is published and well aligned with the Linked Data principles.


**Subreviewer:**

I submitted this review.

**Weak Points:**

- The quality of the resource is not well addressed in the paper. E.g. what is the syntactical/semantical quality?
- The relation to the related work (Section 2) is not clear. E.g. to what extend the created resource is complementary or in what aspects it differs. While this is partially addressed, it is node made explicit. A clear comparison would be very beneficial here.

---

> ### Author Rebuttal · Authors · 2021-01-29
>
> We thank the reviewer for his/her valuable remarks. Below we answer the different concerns raised, and we briefly explain the changes we will make to the camera ready version of the paper, if accepted.
>
> > It is unclear what is the quality of the published data
>
> You are right. Indeed, quality checking can be improved and this is a continuous process. Some metadata coming from external data sources could be erroneous or conflicting. For instance, some Wikipedia pages for a song describe the song itself, but sometimes also many cover versions. In such cases, the infobox (used by DBPedia extractors) could contain metadata for ALL cover versions at the same time.
>
> The hackathons help in spotting many errors, conflicts and recurring problems, and we wrote a set of scripts (available in the git repository) to address some of them. Currently, we are developing visualization tools and inference rules that can be used by the community to report problems (in a crowdsourcing maner), and to better detect conflicting metadata (internally or with other data sources). We will describe this continuous quality process in the paper.
>
>
> > It is not clear to what extend the resource is complementary/differs w.r.t. the other related works/datasets.
> > The relation to the related work (Section 2) is not clear. E.g. to what extend the created resource is complementary or in what aspects it differs. While this is partially addressed, it is node made explicit. A clear comparison would be very beneficial here.
>
> Thanks for this comment. We will add a comparative table to explain the main differences between WASABI dataset and other comparable popular and commercial datasets.
>
> In short, the main existing dataset is the Million song dataset, which was created with music information retrieval methods and it contains very few metadata about the lyrics (only “bag of words” with occurences of words). Moreover, the main source of cultural data  (i.e descriptions of artists, members, location, recording studio, producer, etc.) is MusicBrainz and it contains much less cultural metadata than the WASABI dataset. When comparing WASABI dataset and Million Song dataset, we found that WASABI covers more data sources and more recent data. Furthermore, the lyrics analysis embedded into the WASABI dataset is unique and validated by different methods described in references 2-3 (Fell et alI, 2020).
>
> We will also add comparison elements wrt. Linked Data projects. In particular, the DOREMUS project that overlaps with the WASABI yet in a rather different context as it focuses on classical and traditional music and it does not tackle audio and lyrics processing.
> The Listening Experience Database collects people’s music listening experiences as they are reported in documents like diaries or letters, but consequently mostly relates to legacy music that has little overlap with WASABI. The MIDI Linked Data project publishes a large set of MIDI files in RDF. Linked to DBpedia and relying on the MO, it could complement WASABI to jointly exploit MIDI files and audio and text analyses. Interestingly, some MIDI content was used in WASABI during the evaluation of the chords extraction.
>
>
> > Another minnor comment - Sec 4. It would be nice to add visualisation (figure) of the ontology in the paper.
>
> Indeed. Due to space constraints, we chose to only include an example representation of most typical entities (song, album, artist), and we discussed the ontology in the text. We will consider adding another figure specifically about the ontology.
>
>
> > The quality of the resource is not well addressed in the paper. E.g. what is the syntactical/semantical quality?
>
> For the quality of extracted data the WASABI dataset comes with metadata about lyrics analysis that have been validated using different methods described in the 2-3 related papers (cited in this paper).
> The ontology is mostly a simple taxonomy of types that cannot be subject to a strong logical validation.

---

### Official Review · AnonReviewer3 · 2021-01-17
**Impressive data collection and publication effort on commercial songs, with a few issues however**

**Rating:** 2
**Confidence:** 5

**Review:**

The authors present a massive aggregation of data about commercial songs, gathering metadata from several major existing sources, and applying some advanced content analysis tools on audio and lyrics to augment this metadata.
This well written paper presents the sources selected, the way they were processed and enriched, and then published as a linked data knowledge graph in various ways. It completes this with presentation of current and potential usage scenarios and some explorative user interfaces to showcase the data.


## Potential impact

The authors relate what they have done to other relevant work. Their contribution is original, and frankly said, quite massive in its scope, both in terms of technical depth and breadth.

This said I would have welcomed a bit broader inventory of work related to other music, coming from more "official" cultural projects, as exemplified in the LODLAM community (Linked Open Data for Libraries, Archives and Museums). I am thinking of projects like LinkedJazz, DOREMUS, or some of the LD4L projects on special collections in libraries (that could be about music).

I nonetheless believe this resource could be interesting for the SW community, as a case of applying all kinds of best practices for generating and publishing metadata. Two points that I have liked is that the authors do a very good job at handling content that is not necessarily open, and applying/showcasing automated data generation technology that is not necessarily perfect but provides added value on top of existing metadata.


## Reusability

The usage presented is rather constrained to the context of creation of the data (Wasabi project and its partners), even though it points to a longer interest in the datasets, beyond the project's time.

The visualizations given are use case driven, and interesting and promising, but they still look quite exploratory, i.e. the authors seem to present them without a real need having been identified for them. And I have failed to retrieve them on the general Wasabi site (http://wasabihome.i3s.unice.fr/toolsdemos/) or the one of the dataset itself https://wasabi.i3s.unice.fr/
To me the search scenarios 1 and 4 in section 6 look too close to count as separate scenarios, in terms of potential users (and also solutions to implement them).

The dataset as such is probably going to be interesting for a small part of the community. I am not sure there are many researchers working on that part of "culture". And commercial re-uses could be hampered by the NC and SA clauses of the license: this could be especially harmful for composers and streaming services, which are envisioned as users of the data. That said, for most people interested in the topic, this dataset is going to be a goldmine. And there could be some spillover effect, whereby researchers working on developing/testing some advanced technology could use the dataset as a testbed.

The the dataset is also suitable for extensions, e.g. connecting it to other linked datasets, which would adapt it to new usages. Indeed, The authors have done high-quality data design and documentation. The ontology is simple and the paper and online documentation make it easy to approach.


## Design & Technical quality

The process followed to build the dataset is good. It is very intuitive. And rather well documented. The only issue I found is the role played by the "Animux" data source (in Fig 1 but not in the text): does it even exist?
The way to augment the datasets with several content analysis tools is also quite impressive and will probably be inspiring for many.
The design of the ontology follows the best practices for re-using existing namespaces. It is rather small, and quite modular. I was only puzzled by the creation of new classes wsb:Artist_Person and wsb:Artist_Group, which are equivalent to MO classes.
Finally, the dataset description is available as a DCAT description.

A first serious limitation of the dataset could be its coverage. The dataset is seeded with songs from LyricsWikia, a resource that is now offline. How complete is it? Is it really good, if it has been closed? This could be a problem in terms of relevance of the data for its application domain(s), considering that the process does not seem to add songs from other sources that are used, should they contain more songs. Nothing is said about this possibility, in fact.

A quick investigation also reveals some data quality issues, which are quite typical of such large-scale mashups. For example, the metadata for the "Bad" song [1] has two mo:performer. Michel Jackson, but also "Jackson Jackson". And of course it doesn't look like a correct statement. Following up, there seems to be some problems about "Jackson Jackson", either on the portal page [2] or in the data [3].
It could be that this is inherited from data sources. After all, the metadata for "Bad" in DBpedia [4] is also plagued by dct:subject statements that are not fit with the original definition of this property ("dbc:1987 singles" is not the topic of "Bad", let alone "2012 singles"). But some issues could come from the data integration itself. The only evaluation of data quality (besides the ones of the automatic prediction produced by the advanced content analysis tool) was the "WASABI marathons". Maybe they help, but the presence of this issues connected to a very visible example (on the documentation page) makes one doubt how many data quality issues have been identified, and how many have been fixed.

By the way the published data for "Bad" [5] on the documentation page [6] is not in line with the one served live [1]. There is a second mo:performer on the latter, "Jackson Jackson", which is not on the former.

[1] http://ns.inria.fr/wasabi/song/5714dedc25ac0d8aee4aeeb7
[2] https://wasabi.i3s.unice.fr/#/search/artist/Jackson%20Jackson
[3] http://ns.inria.fr/wasabi/artist/56d8449f53a7ddfc01f96e3a
[4] http://dbpedia.org/resource/Bad_(Michael_Jackson_song)
[5] https://github.com/micbuffa/WasabiDataset/blob/master/ontology/example_song.ttl
[6] https://github.com/micbuffa/WasabiDataset


## Availability

The dataset (and the process to build it) are published and provided with good documentation on github.
The authors provide linked data URIs for the resources it contains, a SPARQL endpoint, a DOI backed by Zenodo, the code and data are provided in CC BY-NC-SA (which is not as open as it could be to reach all its potential users, but which nonetheless qualifies as open).
The authors provide elements on sustainability and maintenance, which I found reasonable enough for an ESWC publication.


## Editorial issues

- in table 2 I think the club for explicitness is wrongly placed: the text says that the predicted explicitness produced 715K annotations, but the club is not positioned on that cell. There's also a mismatch between the Deezer emotion annotations as reported in the table (16K) and in the text (18K).

- in section 4.1 the font used to write down ontology elements varies.

- fig 4 is really hard to read. It should probably be made available on github - and ideally a link to a live demo should be given!

# Post-rebuttal comments

I thank the authors for their answer. Considering the quite honest and insightful answers, and looking at the answers given to other reviewers' comment, I believe the paper could be accepted to ESWC, even if the work is not perfect.

On related work, these are good answer, but I'm wondering whether there is a mismanagement of readers' expectation (at least mine) when reading 'cultural' across the paper. When I read this word, I tend to think of collections that are more traditional, or an analysis of music that puts culture up front (like the Musician relationship networks in Linked Jazz). I understand the authors' focus is narrower, and this is fine. But maybe some editorial trick can be found to better reflect the 'cultural' focus (NB: I believe another reviewer has the same issue!)

The explanation on visualizations is welcome, but I am not sure it really answers my concerns about 'exploratory'. I didn't mean to refer to visualizations for exploring (browsing), but to the fact that the whole endeavour (as the authors recognize) is quite opportunistic and lacking evaluation. Then again, addressing this in the paper would be good.
As a matter of fact I have played with the one at
http://mainline.i3s.unice.fr/eswc2021/WASABI_Visualizations/Jasmine/circle_packing_Re.html
and I really fail to see why (even after selecting a specific platform in the top bar) the circles corresponding to types or genre contain sub-circles of completely equal size for all platforms.
Also, this one
http://mainline.i3s.unice.fr/eswc2021/WASABI_Visualizations/MinhNhatDo/index.html
shows a total number of songs (663253) that is way lower than that of the dataset, even if it doesn't really claim to be based on a subset.
From an editorial perspective I am also a bit frustrated that the new links given in the rebuttal still do not seem to match the ones in Fig 3. Considering that the screenshots are very small I was hoping that I could see these visualisations live. And here is the point that I would like to make clear: I'm nitpicking on all this because I believe this stuff is really cool and promising!


About the coverage, the plans presented sound ok for lyrics. That said, my original point on coverage was rather about the songs identified than about getting the lyrics for all songs. I don't doubt that LyricsWikia "remains the most complete one available" but I am worried it could miss some songs.

The number of songs diminishes in the most recent time, down to 0 in 2018 and afterward according to the SPARQL query
SELECT (count(distinct ?song) as ?song) where {?song a wsb:Song; schema:datePublished ?date FILTER regex(?date, "2018")}

The plans for handling data quality issues seem honest. That said there are really big gaps, which I am not sure could be addressed, e.g. while checking for the most recent songs above, I stumbled in the issue that one third of all songs have apparently no release date:
SELECT (count(distinct ?song) as ?song) where {{?song a wsb:Song; schema:datePublished ?date} UNION {?song a wsb:Song; schema:datePublished ?date}}
only returns 1399135 results. I am quite surprised that the information on date would be missing from the original database(s)!

For the use cases, the clarification is useful but having this in the paper would be good. A scenario being "set with the help of" X does not really indicate clearly that it is only focusing on X (in in fact considering that the features require by scenarios 1 and 4 are quite general, I'd be surprised if they are useful only for 1 and 4).

**Anonymity:**

No, I would like my review to be deanonymized.

**Strong Points:**

- size and coverage of dataset
- follows best practices for publishing data
- follows best practices for ontology re-use
- well-executed inclusion of content analysis results

**Subreviewer:**

I submitted this review.

**Weak Points:**

- possible data quality issues, especially wrt coverage and accuracy. Lack of general data quality evaluation
- quite restrictive license

---

> ### Author Rebuttal · Authors · 2021-01-29
>
> We thank the reviewer for his/her valuable remarks. Below we answer the different concerns raised, and we briefly explain the changes we will make to the camera ready version of the paper, if accepted.
>
> > I would have welcomed a bit broader inventory of work related to other music (...).
>
> There are indeed a number of reference works for which we shall add a comparison in the related works section.
> Some goals of the DOREMUS project overlap with the WASABI yet in a rather different context. It focuses on classical and traditional music (popular songs in WASABI), and does not tackle audio and lyrics processing. However, interesting methods have been used in DOREMUS, e.g. the integration of MIDI resources, recommendation, automatic playlists generation, where a joint exploitation of both datasets could be fruitful.
>
> The WASABI ontology extends the Music Ontology (MO), yet the Performed Music Ontology (part of LD4L) or DOREMUS ontology may be considered if future works need to model more accurately the differences between works, performances, expressions etc.
>
> The Listening Experience Database collects people’s music listening experiences as they are reported in documents like diaries or letters. It mostly relates to legacy music that has little overlap with WASABI.
>
> The MELD framework supports the publication of musicology articles with multi-modal user interfaces that connect different forms of digital resources. Some development could be undertaken to allow musicologists publish articles that would leverage musical data from the WASABI RDF knowledge graph.
>
> The MIDI Linked Data project publishes a large set of MIDI files in RDF. Linked to DBpedia and relying on the MO, it could complement WASABI to jointly exploit MIDI files and audio and text analyses. Interestingly, some MIDI content was used in WASABI during the evaluation of the chords extraction.
>
> > The visualizations (...) look quite exploratory
>
> This point indeed deserves clarification. The overall goal of the visualizations is for users to explore the dataset and communicate findings. Each technique is targeted for a certain user profile (journalist, teacher, musiclogist, etc.), e.g. comparing artists' mutual influences, investigating genre popularity overtime. The infovis gallery is populated with work done mainly by master students since 2020. We are updating the gallery with further techniques while explaining their motivations and use cases.
>
> > A first serious limitation of the dataset could be its coverage. The dataset is seeded with songs from LyricsWikia, a resource that is now offline (...).
>
> LyricsWikia was one of the largest dataset at the time we started WASABI. The 2M songs cover all of the “well known” artist’s production (i.e artists with commercial success). Even for more obscure artists, the dataset remains the most complete one available.
>
> Although LyricsWikia is no longer available, we cannot redistribute the lyrics that are copyrighted material. However, some researchers used our ML models and confirmed that it is possible to get full lyrics from other sources, in particular the commercial MusixMatch service or other websites from where they could scrap the lyrics. We have plans to complete the dataset this way during the next three years. Furthermore, all scripts used to collect metadata are available on the github and shall be reused when we’ll update the dataset.
>
>
> > possible data quality issues, especially wrt coverage and accuracy. Lack of general data quality evaluation.
>
> Indeed some metadata coming from external data sources could be erroneous. In particular, some Wikipedia pages for a song describe the song itself, but sometimes also many cover versions. Despite the hackathons that spotted many recurring problems, we cannot detect all errors. Over at least the next three years, as more metadata will be added, we intend to develop new tools to evaluate the quality of these metadata (i.e visualisations, inferences) and better detect conflicting metadata (internally or with other data sources).
>
>
> >I have failed to retrieve them on the general Wasabi site (http://wasabihome.i3s.unice.fr/toolsdemos/). (...)
>
> We have placed the visualization techniques at http://mainline.i3s.unice.fr/eswc2021/WASABI_Visualizations/ and we will add it to the paper.
>
> > the search scenarios 1 and 4 in section 6 look too close to count as separate scenarios, in terms of potential users (and also solutions to implement them).
>
> These two scenarios are similar in that they both involve searches through the database. Yet they are meant for very different users with different needs that require specific user interfaces: journalists on the one hand, music schools and composers on the other. We will try to clarify this further.
>
> > quite restrictive license
>
> This is correct, but unfortunately inevitable given the licences of the data sources we imported. We’ve worked with Deezer's legal services in order not to infringe other licences.

---

### Decision · Program_Chairs · 2021-02-23

**Decision:**

Accept

**Comment:**

Strong Points:

- Size and coverage of the dataset
- Follows best practice for publishing and reuse
- SPARQL endpoint available
- Well developed data pipeline
- Well written paper

Weak Points:

- Potential data quality issues
- Lacks links to some related ontologies
- Use cases lack some detail
- Restrictive license
- Missing evaluation (not essential for resource paper)

Required Changes:
- Dedicate a section/paragraph and discuss the quality of the dataset.